# Temporomandibular Disk Dislocation Impacts the Stomatognathic System: Comparative Study Based on Biexponential Quantitative T2 Maps

**DOI:** 10.3390/jcm11061621

**Published:** 2022-03-15

**Authors:** Piotr A. Regulski, Jakub Zielinski, Kazimierz T. Szopinski

**Affiliations:** 1Department of Dental and Maxillofacial Radiology, Faculty of Medicine and Dentistry, Medical University of Warsaw, 61 Zwirki i Wigury Street, 02-091 Warsaw, Poland; kszopinski@wum.edu.pl; 2Interdisciplinary Centre for Mathematical and Computational Modelling, University of Warsaw, Krakowskie Przedmiescie 26/28, 00-927 Warsaw, Poland; jziel@icm.edu.pl

**Keywords:** temporomandibular disk, quantitative MRI, T2 maps, biexponential analysis

## Abstract

In this study, we aimed to assess the potential impact of temporomandibular disk displacement on anatomical structures of the stomatognathic system using biexponential T2 magnetic resonance imaging (MRI) maps. Fifty separate MRI scans of the temporomandibular joints (TMJ) of 25 patients were acquired with eight echo times. Biexponential T2 maps were created by weighted reconstruction based on Powell’s conjugate direction method and divided into two groups: the TMJ without (32 images) and with (18 images) disk displacement. The disk, retrodiscal tissue, condylar bone marrow, masseter muscle, lateral and medial pterygoid muscles and dental pulp of the first and second molars were manually segmented twice. The intrarater reliability was assessed. The averages and standard deviations of the T2 times and fractions of each segmented region for each group were calculated and analysed with multiple Student’s *t*-tests. Significant differences between groups were observed in the retrodiscal tissue, medial pterygoid muscle and bone marrow. The pulp short T2 component showed a trend toward statistical significance. The segmentation reliability was excellent (93.6%). The relationship between disk displacement and quantitative MRI features of stomatognathic structures can be useful in the combined treatment of articular disk displacement, pterygoid muscle tension and occlusive reconstruction.

## 1. Introduction

The stomatognathic system is a set of functionally connected anatomical elements comprising the teeth, maxilla, mandible, temporal bone, muscles for mastication, temporomandibular disk and other associated soft tissues. One of its most common pathologies, after caries and periodontitis, is temporomandibular disk displacement [1]. The aetiology of temporomandibular joint disorders is multifactorial and can be associated with inflammatory, infectious, traumatic, congenital or neoplastic causes. This condition is reported to be the most common cause of nonodontogenic orofacial pain and the second most common musculoskeletal disorder and can be characterized by heterogenous symptoms [2,3,4]. Displacement can lead to a limited range of mouth opening and nondental pain in the orofacial region, hindering the everyday life of the affected patients [5,6]. 

The clinical evaluation of patients with temporomandibular joint (TMJ) disorders is used to determine the relationship between disk displacement and complaints originating from other tissues, such as muscle tenderness or tooth sensitivity [7]. However, clinical assessment is only qualitative. Attempts have been made to assess TMJ with surface electromyography [2], cone beam computed tomography [8,9] and magnetic resonance imaging (MRI) [10,11,12]. Therefore, we propose a quantitative method based on biexponential T2 maps for the analysis of the relationship between disk displacement and other stomatognathic structures.

Quantitative T2 mapping is an MRI technique for measuring the transverse relaxation time [13,14]. The T2 times are associated with the presence of water in compartments and the collagen fibre orientation [15]. As the water contents of biological tissues vary, they present distinct T2 times on an MRI. Problems arise when a tissue has more than one compartment. In such cases, biexponential T2 maps, which provide two maps, i.e., one for the short and long T2 components, should be used to capture the subvoxel inhomogeneity and local anisotropy of tissues containing various water compartments [16].

Quantitative MRI studies have shown that monoexponential analysis of the T2 time can be useful for assessing the regeneration and recovery of cartilage, tendons and muscles. Attempts have been made to assess the correlation between disk dislocation and the surrounding tissues on monoexponential maps, however, without significant results (except for retrodiscal tissue) [17,18,19]. In our opinion, this result is associated with the fact that the stomatognathic region consists of anatomical structures that are inhomogeneous in terms of water compartments and collagen structure.

Therefore, the aim of this research was to assess the potential impact of temporomandibular disk displacement on other anatomical structures of the stomatognathic system using biexponential T2 maps. The T2 times of segmented regions in patients with and without disk displacement were compared. Moreover, the reference T2 times and standard deviations of each anatomical structure with no disk displacement are presented. Future studies considering the presence of pathology will be able to refer to the reference T2 times given here.

## 2. Materials and Methods

Fifty MRI scans of the TMJ of 25 patients without TMJ pain or mandibular mobility difficulties were included in this study. The inclusion criteria were as follows: age above 18 years; no metallic foreign bodies or artifacts in the assessed region; and no severe osteoarthritis or signs, such as severe articular surface flattening, bone spurs and osteophytes. Examinations were performed from January 2019 to May 2021. Patients (18 females and 7 males, mean age = 39.7 ± 12.3 years) were positioned in the maximum intercuspation position during imaging (Table 1). The study was approved by the local ethics committee (Bioethics Committee’s reference number: AKBE/85/2021).

The MRI examinations were performed using a 1.5-T MRI unit (Philips Achieva) with an eight-channel head coil (eight -channel SENSE Head coil). The turbo spin echo sequence was employed for T2 map acquisition. The following parameters were set: eight echo times = {13, 26, 39, 52, 65, 78, 91, 104 ms}; repetition time = 2000 ms; eigth slices; field of view = 160 × 160 mm^2^; matrix = 560 × 560 voxels; and average acquisition time = 5 min 32 s.

The examinations were performed separately on the left and right sides, resulting in 50 separate images of the TMJ. MR images were oriented parallel to the plane of the ramus of the mandible and covered the head of the mandible, which was in the field of view. No motion artifacts were found. Patients were positioned in a comfortable, stable position.

The images were divided into two groups: TMJ without disk displacement (32 studies) and TMJ with disk displacement (18 studies). Disk displacement was assessed separately by a radiologist with 25 years of experience and by a dentist with 10 years of experience in maxillofacial radiology. The interrater reliability was measured with Cohen’s kappa coefficient (κ).

Biexponential T2 map reconstruction was performed with a quantitative, weighted reconstruction based on Powell’s conjugate direction method implemented on the VisNow T2 Map Plugin library and available for use with the VisNow platform [20]. The main advantage of this reconstruction method is the use of weights, which largely compensates for differences in the noise level between input signals while preserving the boundaries of uniform regions within an image. 

The reconstruction consisted of two T2 maps (one for short and one for long time components) and two amplitude maps of the short and long time components. Amplitude maps were used to calculate short and long fraction maps as the ratio of the appropriate amplitude to the sum of the amplitudes. Detailed characteristics of the biexponential analysis were provided by Chang et al. [21].

Biexponential T2 maps were tested in terms of the adequacy of reconstruction regarding whether the reconstructed voxels were considered biexponential. The voxel was monoexponential or degenerative when at least one relaxation time component was zero, when the difference between the time components was less than 1 ms, when one amplitude was equal to zero or when the mean squared error of fit (MSE) of the monoexponential reconstruction was less than the MSE of the biexponential reconstruction. In such cases, the voxel was considered to be monoexponential and was not taken into further assessment.

The temporomandibular disk, retrodiscal tissue, condylar bone marrow, masseter muscle, lateral pterygoid muscle medial pterygoid muscle, and dental pulp of the first and second molars were manually segmented by the dentist. The segmentation was performed twice at an interval of two months, with manual contour tracing using the VisNow Plugin Medical library [20] (Figure 1). Segmentation was performed on all slices containing the relevant structure. Due to missing molars, segmentation and further analysis of the pulp was not possible in eight joints. The intrarater reliability was assessed with the intersection-over-union measure. The average of two measurements was taken into consideration in further statistical analysis.

The average and standard deviation of the T2 time and fraction for each segmented region in each group were obtained. Multiple Student’s *t*-tests were performed to compare values in each region between the groups. In order to assess the effect of age, the Pearson correlation coefficients between the T2 components, fractions and age for each structure were calculated. Potential relationships of sex with T2 times and fractions and of missing molars with T2 times were analysed with Student’s t-test. Bonferroni’s correction was used to counteract the multiple comparisons problem (*p* < 0.0125) caused by apossible relation between the short and long T2 components and their fractions. The results with *p*-values less than 0.0125 were significant, and *p*-values less than 0.05 showed a trend toward statistical significance. Normality was checked with the Shapiro–Wilk test. Statistical analysis was performed using Statistica software (Tibco, Palo Alto, Santa Clara, CA, USA).

## 3. Results

The mean and standard deviation of the temporomandibular disk, retrodiscal tissue, bone marrow of the condyloid process of the mandible, masseter muscle, lateral pterygoid muscle and medial pterygoid muscle and pulp of the first and second molars are presented in Table 2. 

Statistically significant differences were observed between the displaced and nondisplaced disk groups in the retrodiscal tissue short T2 component (*p* < 0.0001) short fraction (*p* = 0.0050) and long fraction (*p* = 0.0049), the medial pterygoid muscle short T2 component (*p* = 0.0025) as well as the bone marrow long T2 component (*p* = 0.0030) and short T2 component (*p* = 0.0116), indicating the relationship between disk displacement and quantitative MRI features of stomatognathic anatomical structures. The pulp short T2 component showed a trend toward statistical significance (0.0125 < *p* < 0.0500). Sample T2 maps are presented in Figure 2 and Figure 3. Short and long fraction components are presented in Figure 4 and Figure 5, respectively. The Shapiro–Wilk test confirmed normality in all comparisons.

The correlation coefficients between age and short and long T2 components were not significant for the stomatognathic structures except from the lateral pterygoid muscle short T2 component (r = 0.58, moderate correlation, *p* = 0.0001) and long T2 component (r = 0.45, moderate correlation, *p* = 0.001) as well as the masseter muscle short T2 component (r = 0.44, moderate correlation, *p* = 0.001) and long component (r = 0.39, weak correlation, *p* = 0.004). There were no significant relationships between sex, missing molars and T2 times (*p* > 0.05).

The interrater κ for disk assessment was 1, demonstrating the excellent reliability between raters and the consistency of their ratings across subjects. The segmentation reliability was excellent in terms of the intersection-over-union measure (93.6%).

## 4. Discussion

This study confirms the thesis that, at the same time, the alteration of different structures of the stomatognathic apparatus are present when a disk dislocation is present. Statistically significant results show the relationship between disk dislocation and morphological changes in structures both close to (retrodiscal tissue and condylar bone marrow) and far from (medial pterygoid muscle) the disk and that such changes might have an impact on changes in the pulp of the molars; however, further research is required to confirm statistical significance.

An explanation of this phenomenon might be associated with the function of the analysed regions. Disk dislocation leads to the redevelopment of retrodiscal tissue and its adaptation to new functions (replacing the dislocated disk). This can also increase masticatory muscle tension (which is mostly visible in the pterygoid muscle), thus, causing changes in the fibre orientation and water compartments [1,22]. The effect of increasing tension is transmitted as a force applied to the teeth, causing tooth wear, periodontal space widening and regressive pulp changes. Notably, the T2 time differences between the two groups (with and without disk dislocation) indicated changes in water compartments and collagen matrices, which are the basic elements of the functional structures of the analysed anatomical structures [23].

Juras et al. [24] showed a significant increase in the short relaxation time component in degenerated tendon tissues. An increase of the short T2 time was associated with collagen III synthesis (post-traumatically induced) and the binding of a portion of the free and bound water compartments to collagen fibres and proteoglycan molecules [24,25]. The shifts in the proportion of bound water and collagen III synthesis led to an increase in the short T2 time component [26]. Fukawa et al. showed an increase of the T2 time in degenerated cartilage, which reflects the breakdown of the collagen fibril and increase in water content [27]. Robson et al. found that a reduction in the short T2 component and increase in thelong T2 component were observed in tendinopathy [28]. Therefore, a decrease in the short fraction and increase in the long fraction can be expected in degenerated tendons.

The T2 changes in overloaded stomatognathic structures demonstrated in this paper are similar to those seen in degenerated tendons. The retrodiscal tissue, bone marrow and medial pterygoid muscle showed significant increases in the short T2 time, and the masseter muscle, temporomandibular disk and pulp showed insignificant increases in the short T2 time in patients with disk displacement as in degenerated tendons. This difference in significance of the results between the muscles of mastication might be associated with their different load in joints with and without disk displacement. The load of the medial pterygoid muscle is associated with the contralateral excursion of the mandible which can be affected by disk displacement. The medial pterygoid is the strongest muscle responsible for shifting of the mandible towards the contralateral side [29]. Duman et al. compared the length of masticatory muscles in patients with temporomandibular dysfunction with the length of those muscles in a control group [30]. The most pronounced shortening was observed in the medial pterygoid muscle, which was the most affected muscle. Nascimento at al. showed, in an animal model, metabolic and vascular changes in the medial pterygoid muscle caused by occlusal instability. Chronic stress caused reduced capillary density, increased glycolytic metabolism and morphological changes in the mitochondria of this muscle [31]. 

Therefore, the mechanism of overload of the pterygoid muscle and collagen III synthesis might explain the prolongation of the short T2 time of the stomatognathic structures of patients with disk displacement. However, the complex interplay between significant differences in T2 times or fractions and physical explanations for biological phenomena and the clinical consequences of these differences merits additional research that includes a shorter time echo and multiexponential, quantitative sequences [32]. The complex biexponential behaviour of the tissues can be confirmed in other magnetic field strengths.

The effect of age on the T2 times of the pterygoid and masseter muscles does not alter the results of disk displacement. Neither muscle showed a significant difference in the displacement analysis. However, further research in a larger population is required to dispel doubts about the feasible relationships between disk displacement and T2 times in both muscles in different age groups.

The differences were confirmed for the first time in this study in a quantified manner using biexponentially reconstructed T2 maps. In the available literature, monoexponential analysis has shown a significant difference in the retrodiscal tissue between patients with and without disk dislocation and a lack of significant differences in other stomatognathic tissues [17,33]. For methodological order, we checked the differences in the T2 values calculated with the monoexponential reconstruction for each region in our dataset; the monoexponential reconstructions did not provide significant results (*p* > 0.05, Student’s *t*-tests). The fact that no other statistically significant results were found using the monoexponential fit both in our dataset and in the literature indicates that monoexponential reconstruction does not adequately describe the complex structures of most of the examined tissues. The structural complexity of these tissues requires at least biexponential reconstruction. Furthermore, the monoexponential reconstruction showed a higher MSE compared with the biexponential reconstruction (Figure 6). Attempts have also been made to analyse the pterygoid muscles with diffusion-weighted imaging and texture analysis with MRI [34,35].

This research also has an important clinical implication. The treatment of articular disk displacement should be accompanied by combined treatment of the pterygoid muscle and occlusive reconstruction in a conservative, prosthetic or orthodontic manner; however, further studies are required to confirm this thesis.

The average T2 times of the presented structures can be used as reference values for other studies. In the presence of pathologies, these values will likely differ from the norm presented in this manuscript. According to the T2 map reconstruction method, the values should not differ by and should be independent of the MRI examination parameters. The stability of time values is caused by the employed reconstruction method (Powell’s weighted conjugate direction method), which provides accurate results even for images with a low signal-to-noise ratio.

This study has several limitations. First, a larger studied group could show significance of degenerative pulp changes, which, in our study, presented a trend toward statistical significance. Secondly, our studies were performed on a 1.5T MRI system. As the field strength in MRI affects the spatial and contrast resolution, a similar study performed on a 3T system would supply additional information. In our in vivo studies, no histological confirmation of the structure of the retrodiscal tissue or muscles could be obtained. 

## 5. Conclusions

In conclusion, the retrodiscal tissue, condylar bone marrow and medial pterygoid muscle showed statistically significant differences in the biexponential T2 times between patients with and without disk displacement. The T2 time of the molar pulp showed a trend toward a significant difference between the groups. These results confirmed the morphological impact of disk displacement on stomatognathic tissues. Reference values for long and short T2 times are presented and may be of use in further studies.

## Figures and Tables

**Figure 1 jcm-11-01621-f001:**
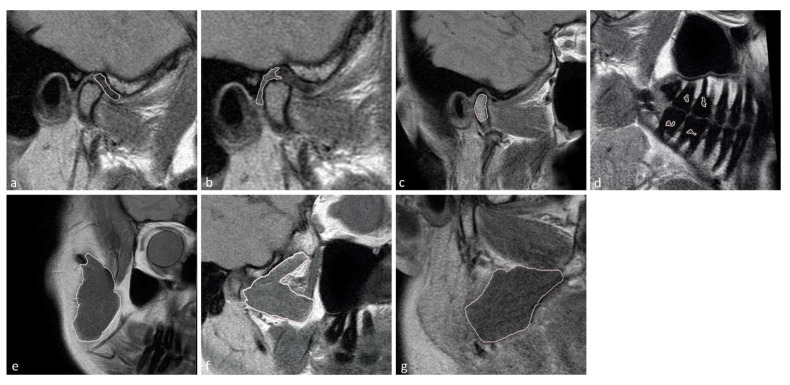
Samples of stomatognathic anatomical structure segmentation for the (**a**) disk; (**b**) retrodiscal tissue; (**c**) bone marrow of the condyloid process; (**d**) pulp; (**e**) m. masseter; (**f**) m. pterygoideus lateralis; and (**g**) m. pterygoideus medialis.

**Figure 2 jcm-11-01621-f002:**
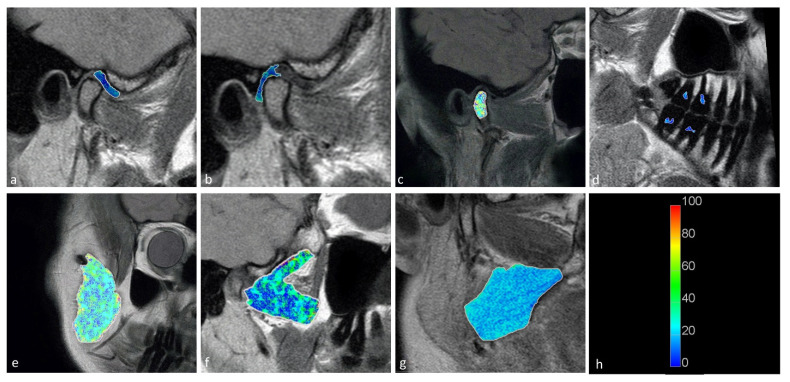
Short T2 maps obtained for the (**a**) disk; (**b**) retrodiscal tissue; (**c**) bone marrow of the condyloid process; (**d**) pulp; (**e**) m. masseter; (**f**) m. pterygoideus lateralis; and (**g**) m. pterygoideus medialis. (**h**) Color scale showing the reference T2 times in seconds.

**Figure 3 jcm-11-01621-f003:**
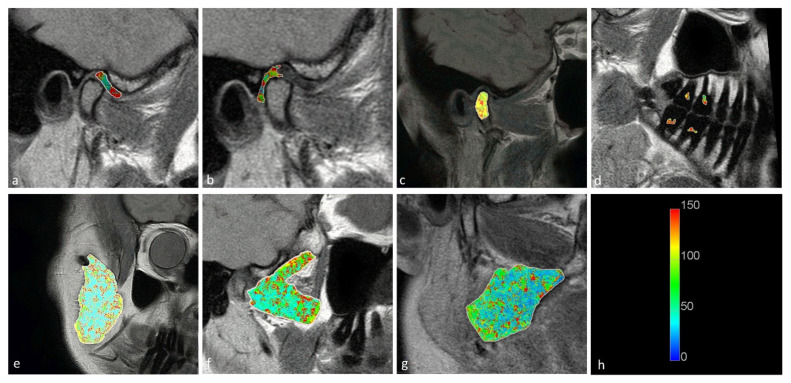
Long T2 maps obtained for the (**a**) disk; (**b**) retrodiscal tissue; (**c**) bone marrow of the condyloid process; (**d**) pulp; (**e**) m. masseter; (**f**) m. pterygoideus lateralis; and (**g**) m. pterygoideus medialis. (**h**) Color scale showing the reference T2 times in seconds.

**Figure 4 jcm-11-01621-f004:**
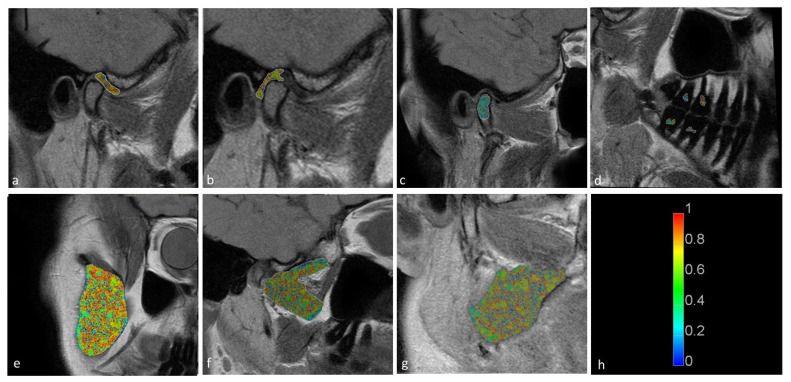
Short fraction components for the (**a**) disk; (**b**) retrodiscal tissue; (**c**) bone marrow of the condyloid process; (**d**) pulp; (**e**) m. masseter; (**f**) m. pterygoideus lateralis; and (**g**) m. pterygoideus medialis. (**h**) Color scale showing the reference values.

**Figure 5 jcm-11-01621-f005:**
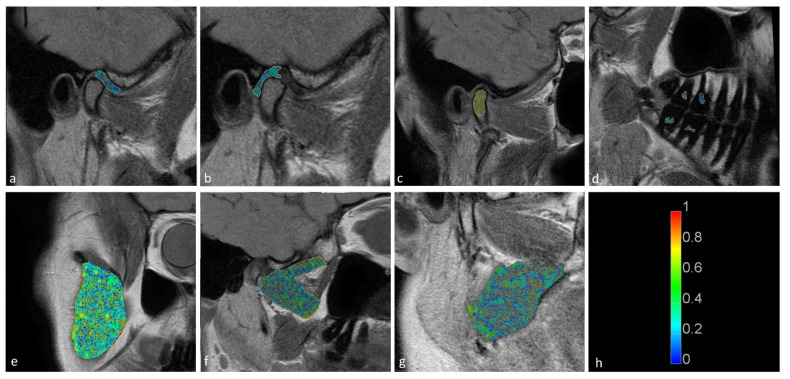
Long fraction components for the (**a**) disk; (**b**) retrodiscal tissue; (**c**) bone marrow of the condyloid process; (**d**) pulp; (**e**) m. masseter; (**f**) m. pterygoideus lateralis; and (**g**) m. pterygoideus medialis. (**h**) Color scale showing the reference values.

**Figure 6 jcm-11-01621-f006:**
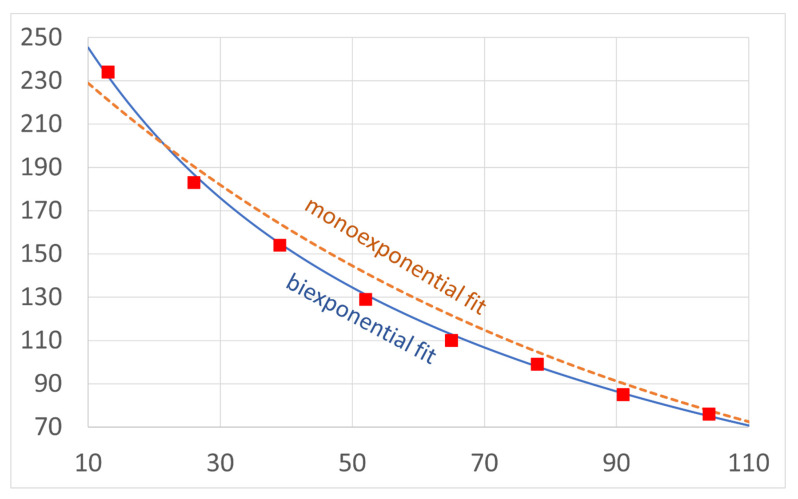
Comparison of mono- (dashed line) and bicomponent (solid line) fitting curves for eight exemplary echo time points (squares) showing a higher error for the monoexponential fit.

**Table 1 jcm-11-01621-t001:** Characteristics of the included studies.

	Disk Displacement	No Disk Displacement	Total No. of Studies
No. studies	18	32	50
Females	14	12	36
Males	4	10	14
Mean age [years]	35.7 ± 13.7	41.5 ± 10.3	39.7 ± 12.3
Females [years]	32.4 ± 11.8	39.4 ± 10.0	36.7 ± 11.3
Males [years]	47.0 ± 13.9	47.6 ± 10.2	47.4 ± 11.4

**Table 2 jcm-11-01621-t002:** Results of the studies. The bolded values show the tendency to significance. The colored and bolded indicate the significant values.

		Disk Displacement		No Disk Displacement		
Anatomical Structure	Component	Mean [ms]	Standard Deviation	Mean [ms]	Standard Deviation	*p*-Value
Temporomandibular disk	short T2	13.3	4.5	12.9	2.1	0.6644
	long T2	105.9	15.5	107.8	16.6	0.6968
	short fraction	0.708	0.043	0.718	0.033	0.2225
	long fraction	0.279	0.043	0.275	0.034	0.1955
Retrodiscal tissue	short T2	22.6	5.2	17.6	3.0	** 0.0001 **
	long T2	105.7	12.4	105.3	12.0	0.8674
	short fraction	0.637	0.039	0.666	0.030	** 0.0050 **
	long fraction	0.359	0.041	0.328	0.032	** 0.0049 **
Bone marrow	short T2	37.8	7.3	42.4	5.3	** 0.0116 **
	long T2	123.7	19.6	114.4	7.4	** 0.0030 **
	short fraction	0.411	0.042	0.392	0.033	0.0832
	long fraction	0.588	0.042	0.607	0.033	0.0775
M. masseter	short T2	25.3	4.8	23.6	3.3	0.1373
	long T2	94.9	11.6	92.0	6.5	0.2621
	short fraction	0.606	0.043	0.614	0.020	0.3476
	long fraction	0.394	0.043	0.385	0.020	0.3544
M. pterygoideus lateralis	short T2	25.9	3.1	26.5	2.6	0.4331
	long T2	98.8	7.9	99.3	6.4	0.8321
	short fraction	0.583	0.027	0.589	0.024	0.4770
	long fraction	0.416	0.027	0.411	0.024	0.4830
M. pterygoideus medialis	short T2	25.3	2.0	22.3	2.1	** 0.0025 **
	long T2	99.5	7.7	97.1	5.0	0.1836
	short fraction	0.604	0.021	0.607	0.019	0.6007
	long fraction	0.395	0.022	0.392	0.019	0.6261
Pulp	short T2	26.8	4.8	21.6	3.2	**0.0499**
	long T2	146.7	16.8	147.0	11.1	0.9515
	short fraction	0.773	0.051	0.489	0.053	0.3388
	long fraction	0.515	0.056	0.495	0.059	0.2844

## Data Availability

Data generated or analysed during the study are available from the corresponding author on request.

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
