# Peer review of "Temporomandibular Disk Dislocation Impacts the Stomatognathic System: Comparative Study Based on Biexponential Quantitative T2 Maps"

_jcm, 2022, doi:10.3390/jcm11061621_

Round 1

Reviewer 1 Report

Excellent study exploring use of Biexponential Quantitative T2 Maps in TMJ disk displacement. An important study to spur further use of such methods in TMJ. Methodology of the study is sound. Results and discussion were presented elegantly. This study would add new evidence of the use of "Biexponential" T2 mapping for TMJ disorder.   

Detailed comment:   Title and abstract: Title and abstract are appropriate and represent the content of the study well. While the use of T2 mapping to assess TMJ tissue changes has been investigated, the use of "Biexponential" T2 mapping is interesting and recent. Introduction:  Introduction was written well. Previous findings on the monoexponential Quantitative T2 Mappping in TMJ was mentioned and  the gap that this study  is trying to fill  and the study aim were succinctly explained.   Methodology: Methodology of the study is sound scientifically and appropriate for the aim of the study. Steps were explained clearly.  Results: Findings of the study were presented concisely. Figures provided were of high quality and aided the presentation of results. Discussion and summary: The results' significance were discussed in detail. Findings that might be controversial (involvement of pulp changes) were logically associated with clinical possible situations. The conclusion answered all the objectives of the study.

Author Response

Thank you for your kind and comprehensive review.

Reviewer 2 Report

This paper reported that the effect of with or without articular disc dislocation using T2 values of the articular disc, retrodiscal tissue, condylar bone marrow, masseter muscle, lateral pterygoid muscle, medial pterygoid muscle, and dental pulp. Then, the results indicated that retrodiscal tissue, molar bone marrow, and medial pterygoid muscle may be affected in patients with disc displacement. However, the author mentioned that T2 value calculated by the monoexponential reconstruction did not provide significant results. So T2 value from biexponential reconstruction method may be more sensitive method comparing to the monoexponential reconstruction method. These were the reasonable results for me. However, even if this sensitive method was used, the results could not be shown for the masseter muscle, lateral pterygoid muscle, and articular disc, which are considered to be affected. I believe this is due in part to the very small sample size. This result was also reasonable for me. Therefore, I think that the impact of this paper on the readers is very significant.

Some specific comments were shown as below.

Title; “Biexponential, Quantitative T2 Maps” Do you need comma between Biexponential and Quantitative?

P3L105; The segmentation was performed twice at an interval of two months,

Is the T2 value you indicated in this paper the average of two measurements? Or is it one of the measurements? Please clarify it.

P3L108; Due to missing molars, segmentation and further analysis of the pulp was not possible in eight cases.

Eight cases mean eight people, right? Or eight joints? If it is eight people, it means that 32% of the subjects were excluded. I think this exclusion rate is too high and should avoid referring to the effect of the pulp on these 25 total cases.

Table 2; The author showed four value of short T2 component, long T2 component, short fraction, and long fraction. The author should discuss these four differences and their relation to biological phenomena.

Reviewer 3 Report

The study presents an investigation of differences in T2 maps for different tissues of the stomatognathic system in disc dislocation patients. Generally, the paper is well written and addresses an interesting and important question. There are some methodological questions that need to be addressed.

General comments

  • The authors state that they used a 1.5T MRI machine. As far as I am aware 3T is the standard for T2 Maps and even higher field strengths are used. For biexponential behavior, literature suggests that higher field strengths are even more important to accurately detect differences [1]. How do the authors argue against that concern?

  • The authors state multiple times that “Fifty separate MRI scans of the temporomandibular joint (TMJ) of 25 patients 13 were acquired”. Does this mean that one TMJ was scanned two times or did both TMJs get scanned once?

  • I think the paper is lacking better explanations for the differences in T2 values found for most tissues. Especially, the difference in the medial pterygoid muscle needs more explanation. Only one of the four categories gives a statistically significant difference. I do not follow the explanation given. Why would disc dislocation increase medial pterygoid activation and not masseter activation? How would you see changes in tension when the mouth is in a “relaxed” closed-mouth position? Also, both references cited with the explanation do not talk about muscles at all [2] or only talk about the lateral pterygoid muscle and still don’t give the same explanation [3].

Specific comments:

Line 63-64: Why were patients with TMJ pain and mandibular mobility difficulties excluded?

Line 92-93: What was the cut-off between short and long-time components?

Line 93-94: The word “amplitude map” is not mentioned in the whole manuscript except for this one sentence, are these the maps of the short and long fraction components? I do not fully understand how these were created and separated? Could you explain the process and the reasoning behind these maps in more detail?

Table 2: I’m curious about the lack of differences in the TMJ disc? Wouldn’t you expect the displaced disc to have different levels and directions of mechanical load applied and hence see more changes?

Line 175-176: While I don’t necessarily disagree with the notion that malocclusion could influence the TMJ, I don’t see where this is shown in the study?

Line 186-191: Not all structures mentioned in this sentence show a significant difference in your study.

Line 214-216: Again, while I don’t necessarily disagree with your sentence, I don’t think your study presents enough evidence to support such a definitive statement.

Line 217-223: As far as I am aware it has been shown that T2 values differ significantly between field strengths and even machines at the same field strengths. More recently even angles of the acquisition have been shown to influence the signal [4]. Can you please explain in more detail how your reconstruction method solves all these problems?

References:

[1]         J. A. Rioux, I. R. Levesque, and B. K. Rutt, “Biexponential Longitudinal Relaxation in White Matter: Characterization and Impact on T1 Mapping with IR-FSE and MP2RAGE,” Magn. Reson. Med., vol. 75, no. 6, p. 2265, Jun. 2016.

[2]         C. Lee, Y. J. Choi, K. J. Jeon, and S. S. Han, “Synthetic magnetic resonance imaging for quantitative parameter evaluation of temporomandibular joint disorders,” Dentomaxillofacial Radiol., vol. 50, no. 5, pp. 1–6, 2021.

[3]         D. Manfredini, “Etiopathogenesis of disk displacement of the temporomandibular joint: A review of the mechanisms,” Indian J. Dent. Res., vol. 20, no. 2, p. 212, Apr. 2009.

[4]         B. Hager et al., “Transverse Relaxation Anisotropy of the Achilles and Patellar Tendon Studied by MR Microscopy,” 2022.

Reviewer 4 Report

I carefully read the article entitled: “Temporomandibular Disk Dislocation Impacts the Stomatognathic System: Comparative Study based on Biexponential, Quantitative T2 Maps”.

I think the article is interesting but requires a number of adjustments.

Point by point these are my tips.

INTRODUCTION

This section should be supplemented with brief references to TMD, TMD diagnosis and diagnostic tests, so that you can focus the attention on bioexponential T2 maps.

I invite you to consider the following article in order to complete the introduction:

Doi:10.1097/SCS.0000000000004376

MATERIAL AND METHODS

The group investigated is limited by number and heterogeneous by age, however TMD occurs at different ages and with a different clinic.

Reduce the length of some periods.

Page 2 line 80 - What does it mean : “… and covered the head of the mandible” ?

Page 2 line 69 – Write only: (Table 1).

Page 2 line 81/82 – Write only once. See line 68.

Page 3 line 96/97 – The text isn’t very clear.

Describe the statistical analysis more accurately.

DISCUSSION

You cannot write that: “This study confirms the thesis that disk dislocation has an influence on other structures of the stomatognathic system”.

You can say that at the same time the alteration of different structures of the stomatognatic apparatus are present when a disk dislocation is present.

There’s no cause and effect ratio.

TMD ‘s etiology is multifactorial and related to a lot of factors.

At this point I don’t think the title of this paper is correct.

Therefore, also the conclusions have to be revised.

Highlight the CONCLUSIONS section

Your study deserves to be deepened and expanded.

I wish you good work.

Round 2

Reviewer 2 Report

The author made good revision for all reviewers’ comments. However, there is one mistake.

P10L240; with and without disc displacement.

The author used “disk” in the all text, but used “disc” in this sentence. Please unify it.

Author Response

The author made good revision for all reviewers’ comments. However, there is one mistake.

P10L240; with and without disc displacement.

The author used “disk” in the all text, but used “disc” in this sentence. Please unify it.

Response 1: The misspelling was corrected (P7L208).

Reviewer 3 Report

The authors have adequately answered all my comments. I only have one minor suggestion left:

I'd suggest changing "Fifty separate MRI scans of the temporomandibular joint (TMJ) of 25 patients 13 were acquired with eight echo times." in the abstract to the plural "Fifty separate MRI scans of the temporomandibular joints (TMJ) of 25 patients 13 were acquired with eight echo times." to make it clearer that both TMJs were investigated.

Author Response

The authors have adequately answered all my comments. I only have one minor suggestion left:

I'd suggest changing "Fifty separate MRI scans of the temporomandibular joint (TMJ) of 25 patients 13 were acquired with eight echo times." in the abstract to the plural "Fifty separate MRI scans of the temporomandibular joints (TMJ) of 25 patients 13 were acquired with eight echo times." to make it clearer that both TMJs were investigated.

Response 1: The sentence was changed according to the Reviewer’s suggestion.